# Label-Free Direct Detection of Saxitoxin Based on a Localized Surface Plasmon Resonance Aptasensor

**DOI:** 10.3390/toxins11050274

**Published:** 2019-05-15

**Authors:** Su-Ji Ha, Jin-Ho Park, Bobin Lee, Min-Gon Kim

**Affiliations:** Department of chemistry, school of physics and chemistry, Gwangju Institute of Science and Technology (GIST), 123 Cheomdangwagi-ro, Buk-Gu, Gwangju 500-712, Korea; suji1420@naver.com (S.-J.H.); jhp1223@gist.ac.kr (J.-H.P.); bb7896@gist.ac.kr (B.L.)

**Keywords:** marine-toxins, saxitoxin, LSPR, aptamer, aptasensor

## Abstract

Seafood is an emerging health food, and interest in improving the quality of seafood is increasing. Saxitoxin (STX) is a neurotoxin produced by marine dinoflagellates that is accumulated in seafood. It can block the neuronal transmission between nerves and muscle cell membranes, resulting in the disturbance of neuromuscular transmission and subsequent voluntary muscle paralysis. Here, we developed a new aptamer for the detection of STX using graphene oxide–systematic evolution of ligands by exponential enrichment (GO-SELEX). Furthermore, we confirmed sensitivity and selectivity of the developed aptamer specific to STX using a localized surface plasmon resonance (LSPR) sensor. The sensing chip was fabricated by fixing the new STX aptamer immobilized on the gold nanorod (GNR) substrate. The STX LSPR aptasensor showed a broad, linear detection range from 5 to 10,000 μg/L, with a limit of detection (LOD) of 2.46 μg/L (3σ). Moreover, it was suitable for the detection of STX (10, 100, and 2000 μg/L) in spiked mussel samples and showed a good recovery rate (96.13–116.05%). The results demonstrated that the new STX aptamer-modified GNR chip was sufficiently sensitive and selective to detect STX and can be applied to real samples as well. This LSPR aptasensor is a simple, label-free, cost-effective sensing system with a wide detectable range.

## 1. Introduction

Marine biotoxins accumulate in seafood through the aquatic food chain [1,2]. They cause various types of poisoning, such as amnesic shellfish poisoning (ASP), diarrhetic shellfish poisoning (DSP), and paralytic shellfish poisoning (PSP), depending on the type of toxin [3]. Toxicity can occur as a result of ingestion of contaminated seafood, inhalation, or contact on skin; further, most marine toxins cannot be destroyed under high temperature and acidic conditions. Specifically, PSP toxins are potent neurotoxins produced by marine dinoflagellates [4,5]. Exposure to PSP results in memory loss, disorientation, nausea, vomiting, and paralysis. Therefore, the development of detection methods and continuous monitoring are needed to prevent the harvest and consumption of PSP-contaminated seafood.

Saxitoxin (STX), which is the most representative PSP, binds to the voltage-gated Na^+^ channel of neurons and blocks neuronal transmission between nerves and muscle, and this can cause fetal illness, e.g., cardiovascular shock, respiratory arrest, and even death [5,6,7]. For these dangers, a guideline of 75 ug/kg STX in seafood has been suggested by the European Food Safety Authority (FESA) [8]. Several methods are available for the detection of STX. The mouse bioassay is a standard method established by Sommer and Mayer in 1937, according to the Association of Analytical Communities (AOAC) [9]. However, this method has some drawbacks, such as low sensitivity and accuracy, it requires more labor, and also presents ethical problems [10,11]. Therefore, in order to replace this method, high-performance liquid chromatography-fluorescence (HPLC-FL) and -mass spectrometric (HPLC-MS) have been utilized [10,11,12,13,14,15]. These methods offer highly sensitive and accurate analytical performance; however, they also have disadvantages such as complex operating preprocesses, high cost, and requirement of professional operators. Therefore, simple, cost-effective, and efficient methods for monitoring of PSP toxins have been developed using diverse sensing platforms, e.g., lab-on-a-chip, surface plasmon resonance (SPR), electrochemistry, and fluorescence assay [16,17,18,19,20,21]. However, these PSP-sensing tools still face limitations, e.g., a relatively narrow linear detection range and decreasing signal-based analysis. In the case of the observation of the signal change that decreases from the control condition, the unstable initial signal degree at the control condition may be a critical problem. Thus, this method is less quantitative unlike an increasing signal-based analysis, where the starting point is a zero signal [21]. 

For the detection of toxins with increasing signals, our group developed a localized SPR (LSPR) aptasensor by integrating a gold nanorod (GNR) and the aptamer [22,23,24,25]. Basically, the LSPR-based sensor typically uses intrinsic electric field waves of novel metal nanoparticles (NPs), which are sensitive to changes in the refractive index (RI) value around the NPs [26,27,28,29]. The molecular-binding events near the surface of the NPs induce a wavelength shift in the absorption spectrum of NPs, which is called the LSPR shift, and target molecules can be quantitatively determined by measuring the amount of LSPR shift. In the previous studies of our group, the structural change in aptamer by target/aptamer binding event on the GNR surface induces a large RI increase, and this performance eventually generates a strong LSPR shift. Then, the target molecule can be recognized by measuring the amount of LSPR shift. Actually, various small molecules (<900Da), such as mycotoxins (OTA, AFB1), biomolecule (ATP) and chemical ion (K^+^), could be quantitatively determined. So, a key mechanism is the conformational change of aptamer by the target molecules.

However, until now, STX aptamers that undergo structural change caused by the target have not been reported; it is known that a geometric change in the aptamer does not occur by capturing STX; or the underlying mechanism has not been clearly defined [30,31]. Thus, we tried developing an STX aptamer that can induce a geometric change with the target. Regarding this, a graphene oxide-the systematic evolution of ligands by exponential enrichment (GO-SELEX) was chosen for STX aptamer development, because GO-SELEX can provide more possibility for conformational change with target/aptamer complex comparing to ordinary SELEX. In brief, contrary to SELEX where targets are fixed on a solid substrate, a random library of DNA is attached on the GO surface. When target treated to the mixture of DNAs and GO, DNA sequences having specificity against a target are automatically released from the GO surface to capture the target. At the target-capturing event, the aptamer can have relatively more chances to be changed structurally compared to the SELEX because there are no/or very tiny steric hindrance and spatial restriction due to the freely-floating targets in solution. Indeed, the GO-SELEX has been developed to overcome the inherently negative influence of the immobilized target to aptamer property of the SELEX.

## 2. Results

### 2.1. Development of a New STX Aptamer

#### 2.1.1. GO-SELEX for STX Aptamer 

Aptamers are commonly developed using the SELEX. The concept SELEX was invented by Tuerk et al. in 1995, and this technique finds specific sequences with high affinity to a target material. Since the SELEX was developed, professor Gu in Korea University introduced an improved method for screening aptamers, called the GO-SELEX [32]. This aptamer development method shows high efficiency and yield because the π-π stacking interaction between GO and single-stranded DNA (ssDNA) is utilized, unlike in the old method where the target is fixed on the solid substrate. In addition, easy handling is possible because the detached complexes are automatically separated from GO. The aptamer-developing process used in this study is shown in Scheme 1.

The random sequence library (ssDNA pool) was incubated with GO in the working buffer for 1 h. The ssDNA was adsorbed onto the GO surface through π-π stacking and hydrophobic interaction [33,34,35]. After incubation, the target molecule was added to the solution. Then, sequences that can bind to the target with unique three-dimensional (3D) structure were broken off from the GO (target/aptamer complexes are released into solution in this step). Next, the solution was centrifuged to remove the residue, and the supernatant was recovered and amplified. The amplified ssDNA was used for the next round. The counter selection was also performed using the gonyautoxin (GTX) complex that has a similar chemical structure to the STX. First, the counter target was incubated with the random sequences and GO, sequentially. After centrifugation, the supernatant was discarded and the residue was suspended to repeat the positive selection cycles.

#### 2.1.2. Characterization of the STX Aptamer

In Appendix A, electrophoresis gel images of the GO-SELEX rounds are shown. A uniform band of random library DNA (76bp) was confirmed in each gel image. The quantity of DNA obtained from each selection cycle was measured using the NanoDrop spectrophotometer, and an increasing tendency of DNA concentration was recorded along with selection cycles (Figure 1A). The selection cycles were stopped after the ninth cycle, then, the enriched ssDNAs were cloned and sequenced to be “TAGGGAAGAGAAGGACATATGATGGCACAAGGCCTCATCAATCGGTATACGGGTTGACTAGTACATGACCACTTGA”. As shown in Figure 1B, the secondary structure of the sequence that has the lowest △G was predicted by the Mfold web server. 

Fluorescence assay and circular dichroism (CD) spectra analysis were performed to confirm the affinity and STX-binding performance of the aptamer. First, the fluorescence assay was performed to confirm the affinity of the aptamer by measuring the difference in fluorescence intensities between different concentrations of the aptamer. The fluorescent aptamer (fluorescein amidite (FAM)-modified at 5’-end of DNA sequence) was incubated with STX and GO. After incubation, the fluorescence of the sample supernatant was measured. To observe the negative control signal, samples without STX were used. As shown in Figure 1C, the dissociation constant (K_d_) value of the STX aptamer was recorded to be 50.75 ± 14.97 nM by calculating the binding saturation curve. 

Second, CD spectra analysis was performed to check the STX-binding performance of the aptamer. The asymmetric backbone sugars of DNA and the helical arrangement appear as fingerprint peaks in the CD spectrum oligonucleotide sequence [36]. As shown in Figure 1D, the CD spectrum of the aptamer (1 μM) was changed by STX (3 μM). After the addition of STX, the intensity near 215 nm was increased and the shape of the spectrum was changed. These changes in CD patterns typically indicate conformational change in the aptamer due to an interaction between STX and the aptamer.

### 2.2. Application to LSPR Aptasensor

As mentioned above, an LSPR aptasensing system for small molecule detection has been developed in our group [22]. In the LSPR aptasensor, a movement of the GNR absorption band to longer wavelength region by RI change on GNR surface originating from the geometric variation between aptamer and target/aptamer complex. This phenomenon is called an LSPR shift, and target molecules can be quantitatively recognized by measuring the amount of LSPR shift. 

The STX aptamer (M-30) was first reported in 2013 [36], and a sequence screening study of the aptamer was conducted through mutagenesis and truncation [37]. Among the screened sequences, M-30f showed the highest binding affinity against STX. However, it was reported that the geometric switch of the aptamer does not occur by binding to the STX [31]. Considering that a conformational change of the aptamer is effective for the LSPR aptasensor (it induces a large increase in molecular density within sensing distance of the NP), the newly developed STX aptamer seemed more suitable to the LSPR aptasensing platform than the previously reported aptamer, due to its ability to induce structural change. Therefore, the LSPR sensing chip was fabricated by immobilization of the new STX aptamer on the GNR substrate (Figure 2A). Then, STX could be captured by an aptamer-modified GNR substrate, and the quantification of STX was possible by measuring the maximum wavelength shift in the LSPR spectra.

#### 2.2.1. Optimization of the Oligonucleotide Molar Ratio

STX has a positive charge in above-neutral pH buffers. Therefore, we hypothesized that there could be an electrostatic interaction between STX and the aptamer. The working buffer with the least non-specific adsorption was first screened with LSPR sensing chips fabricated by fixing only poly T_3_ on the GNR substrate. The results showed the change in LSPR shifts depending on several working buffers without STXs (Appendix A). The non-specific signal was relatively small at low pH. Therefore, all the following tests were conducted in the citrate buffer (pH 3.0). 

For the geometric change of aptamers to a unique 3D structure, enough spaces among the aptamers are necessary. In the case of a highly dense fixation of aptamers on the GNR surface, their own folding may disrupt the conformational change of the aptamers. Therefore, a spacer is needed to ensure sufficient space to enable a conformational change. In this study, poly T_3_ that consists of three thymines was used as a spacer, and the molar ratio between aptamer and spacer was optimized. The LSPR aptasensor substrates were individually prepared with different molar ratios of oligonucleotides (aptamer:poly T_3_ = 0:1, 1:1, 1:2, 1:3, 1:4, 1:5, 1:6, and 1:7), and then, STX was loaded to each sensing chip with a working buffer (20 mM citrate pH 3.0 with 100 mM NaCl, 10 mM KCl, and 1 mM MgCl_2_). In each condition, average control/positive LSPR signals were recorded as 1.5 ± 0.07 nm/2.83 ± 0.35 nm, 1.83 ± 0.61 nm/3.5 ± 0.55 nm, 1.5 ± 0.17 nm/3.5 ± 0.55 nm, 1.5 ± 0.11 nm/2.67 ± 0.87 nm, 0.17 ± 0.87 nm/3.00 ± 0.95 nm, 1.17 ± 1.95 nm/2.33 ± 0.23 nm, and 1.17 ± 0.29 nm/1.00 ± 0.87 nm, respectively. Among these results, the 1:5 ratio (0.17 ± 0.87 nm/3.00 ± 0.95 nm) showed the most significant difference in the LSPR shift between the presence and absence of STX with the smallest control signal (Figure 2B).

#### 2.2.2. Screening of Working Buffer Components

Monovalent and divalent cations affect the folding topology, stabilization of the aptamer by neutralizing its negative charge, and interaction between the target and aptamer. [38,39,40,41]. Therefore, we checked the effect of the salts used in the working buffer (Appendix A). In the case of monovalent ions, Na^+^ and K^+^, the LSPR shifts were most pronounced at 50 mM and 10 mM, respectively. The LSPR shifts of divalent ions, Mg^2+^ and Ca^2+^, were considerable at 1 mM and 2 mM, respectively. Based on these results, a further screening of salt combinations in the working buffer was performed with optimized concentrations of each ion (50 mM Na^+^, 10 mM K^+^, 1 mM Mg^2+^, and 2 mM Ca^2+^). In the case of the positive LSPR signals (this means that LSPR shift occurs with STX), the combination A (K^+^ and Ca^2+^), D (Na^+^ and K^+^), and E (Na^+^, K^+^, and Ca^2+^) showed relatively large degrees of LSPR shift, and the control LSPR signals (this means that LSPR shift occurs without STX) of the combination A, D, and E were similar (Figure 2C). However, considering the standard deviation (SD) value, the positive signal of the combination E was the most stable. Thus, the following test was performed using the combination E (50 mM Na^+^, 10 mM K^+^, 2mM Ca^2+^). In addition, as shown in Figure 2D, the pH of the working buffer was optimized under the screened salt condition. Among several pH conditions, the largest difference between the control and positive signal and the most stable LSPR shift degrees was recorded at pH 3.0.

### 2.3. STX Detection Using the LSPR Aptasensor

#### 2.3.1. Quantitative and Selectivity Analysis of STX

The target-recognition performance of the LSPR aptasensor with the developed aptamer was tested under the optimized conditions. When STX was added to the sensing system, there was an increase in RI value near the GNR surface due to the STX/aptamer binding event. This eventually caused an increase in the GNR absorption band to the longer wavelength region. In fact, a gradual LSPR shift was observed with the increase in STX concentration from 0 to 10,000 μg/L (Figure 3A). By measuring the amount of LSPR shift, a quantitative analysis of STX was possible. As shown in Figure 3B, for 5, 10, 20, 50, 100, 200, 500, 1000, 2000, 5000 and 10,000 μg/L STX, the degrees of ΔLSPR shift were recorded as 0.50, 0.64, 0.91, 1.14, 1.37, 1.54, 1.77, 2.04, 2.33, 2.74, and 2.90 nm, respectively. A broad linear dynamic range for STX detection was determined from 5 to 10,000 μg/L with good linearity (R^2^ = 0.9896), and the limit of detection (LOD) was calculated to be 2.46 μg/L—the SD value of control condition was 0.2045, and the 3 × SD value (3σ = 0.6135) was applied to the equation, y = 0.3204ln(x) + 0.3248, which is obtained from quantitative analysis of standard STX.

#### 2.3.2. Selectivity Test

The selectivity of the LSPR aptasensor was explored using a GTX complex mixture of GTX 1 and 4 (GTX 1&4) and a mixture of GTX 2 and 3 (GTX 2&3); neoSTX and okadaic acid (OA), which are analogs of the STX (Appendix A), were also used in this study. The GTX complex and neoSTX are neurotoxins in paralytic shellfish toxins and have similar chemical structures to the STX [5]. OA is the representative mycotoxin of the DSP [42]. These different toxins were individually added to the developed sensing system, and the LSPR shifts were compared (Figure 4). When the four samples, i.e., GTX 1&4, GTX 2&3, neoSTX, and OA, were subjected to LSPR aptasensors, only negligible signal changes were observed, while a distinguishable LSPR shift was obviously recorded with STX.

#### 2.3.3. Spiking Test for Real Sample Analysis

In order to verify that the LSPR aptasensor can detect STX in the real samples, we analyzed STX-spiked mussel samples (Table 1). STX was added to homogenized mussel samples, and the mixtures were diluted to 1/100 volumetric ratio in the working buffer to obtain the STX-contaminated real samples having final concentrations of 10, 100, and 2000 μg/L for the spiking test. Although a large LSPR shift of 18.625 nm occurred in the control condition, which is a non-STX-contaminated mussel sample (it is assumed that the strong background signal originated from the matrix substances), the relative LSPR shift recorded by STX-spiked samples with final concentrations of 10 μg/L, 100 μg/L, and 2000 μg/L, was 0.63 nm, 1.59 nm, and 2.24 nm, respectively. The recoveries in the real samples were calculated as 96.13–116.05%, which showed good agreement with the results obtained for standard STX samples. Based on these results, we prospect that the proposed LSPR aptasensor has applicability to be utilized in on-site STX determination.

## 3. Conclusions

This work is the first report of STX detection using an LSPR aptasensor. We developed a novel STX aptamer via GO-SELEX, and this newly-discovered aptamer was integrated with the LSPR sensing platform. The K_d_ value of the STX aptamer was 50.75 ± 14.97 nM, and the structural variation caused by the target molecule, STX, was confirmed by the change in the CD patterns.

Aptamers are considered suitable for fabricating LSPR aptasensors, because the conformational change in aptamer structure due to a target-aptamer binding event is a key mechanism of the LSPR aptasensing system. Regarding this, we could utilize the developed STX aptamer as a receptor in the LSPR aptasensor. Indeed, a GNR absorption band movement to a longer wavelength region was induced by STX/aptamer binding performance due to the change in the surrounding RI value caused by the conformational change in the aptamer on the GNR surface. This LSPR aptasensor showed a broad detection linear range from 5 to 10,000 μg/L with a high sensitivity determined by an LOD value of 2.46 μg/L. These sensing performances are sufficient for a STX sensor, because EFSA suggests that STX should remain in seafood below 75 μg/kg to ensure safety [8]. Moreover, the proposed LSPR aptasensor was applied to real sample analysis. STXs in real mussel samples were successfully determined with good recoveries (96.13–116.05%). It is expected that the developed sensing system integrated with the newly-discovered STX aptamer can be utilized for real-field STX analysis with high accuracy and selectivity.

## 4. Discussion

Furthermore, the sensing abilities of several STX detection methods reported previously and that of the developed LSPR aptasensor were compared (Table 2). The LOD obtained from this work was relatively high, however, it surely meets the EFSA guideline (~75 μg/kg). The recovery estimated in the real sample analysis was comparable with that in other methods. Three points are of particular interest. First, is the increasing signal-based measurement. Second, is the widest dynamic range (5 to 10,000 μg/L). Third, is the process after sample pre-treatment comprises very simple measurement steps (only two steps).

However, our proposed sensing method shows a small LSPR signal (~3.5 nm), and this causes a need for the spectrometer to have a relatively high-resolution (≤1 nm) to distinguish an LSPR shift depending on concentrations of the target. In addition, the dilution of a real sample is required to reduce the matrix effect which causes a non-specific LSPR shift. So, in future work, we will study strategies for signal enhancement and improved pre-treatment procedure of real samples.

Although some disadvantages have been described above, we believe that our research can offer useful insights that impact broad interdisciplinary research fields as the first report of STX determination based on an LSPR aptasensor and its high-performances: high selectivity and sensitivity, broad quantification of STXs, and application to a real sample test.

## 5. Materials and Methods

### 5.1. Materials and Reagents

STX was obtained from NIST (Maryland, USA). GTX complex and neoSTX were obtained from CIFGA (Lugo, Spain). Okadaic acid, graphene oxide, sodium borohydride (NaBH_4_), gold (III) chloride trihydrate (HAuCl_4_), silver nitrate (AgNO_3_), L-ascorbic acid, cetyltrimethylammonium bromide (CTAB), 3-mercaptopropyltrimethoxysilane (MPTMS), potassium chloride, sodium chloride, calcium chloride, sodium citrate, citric acid, acetic acid, sodium acetate, hydrochloride acid, and tris(2-carboxyethyl) phosphine hydrochloride (TCEP) were obtained from Sigma Aldrich (St. Louis MO, USA). In addition, 1X phosphate buffered saline (PBS) buffer and Tris buffer were obtained from Biosesang. Pfu Plus DNA polymerase and 200 μM dNTP were obtained from Elpis (Daejeon, Korea). Agarose gel was obtained from Roche (Basel, Switzerland). Lambda Exonuclease and DH5α cells were obtained from Enzynomics (Daejeon, Korea). Random sequences were custom-synthesized by IDT (Coralville, IA, USA). Forward primer (5’-TAGTAGGGAAGAGAAGGACATAT-3’), reverse primer (5’-TCAAGTGGTCATGTACTAGTCAA-3’), STX aptamer (5’-TTTTTTAGGGAAGAGAAGGACA TATGATGGCACAAGGCCTCATCA ATCGGTATACGGGTTGACTAGTACATGACCACTTGA-3’), and poly T_3_ (5’-SH-TTT-3’) were obtained from Mbiotech (Hanam, Korea) and GENOTECH (Daejeon, Korea). Mussel samples were collected from an online market in Korea in 2017. The edible portion of the mussel samples was homogenized and stored in a deep freezer (−80°C) until analysis.

### 5.2. In vitro Selection of the DNA Aptamer

#### 5.2.1. Graphene Oxide-SELEX Process

For the development of an aptamer, we performed GO-SELEX (Scheme 1) [33,35]. Briefly, 1 μL of random sequence library (stock 3 μM) was dissolved in 1X phosphate buffered saline (PBS) buffer and mixed with 30 μL GO (stock 2 mg/mL). The mixture was incubated with rotation at room temperature (RT) for 1 h, and 4 μL of STX (stock 100 ppm) and 6 μL of methyl alcohol were added to the mixture. The mixture was incubated with rotation at RT for 1 h. After incubation, the mixture was centrifuged at 15,000 rpm for 20 min and the supernatant was collected, purified, and amplified.

The counter selection rounds were performed during the previous 4^th^ and 7^th^ positive selection rounds. Briefly, the counter target (final concentration 4 mg/L) was incubated with a random sequence library in 1× PBS buffer. The mixture was incubated with rotation at RT for 1 h, and 4 μL of GO (stock 2 mg/mL) was added to the mixture. The mixture was incubated with rotation at RT for 1 h. After incubation, the mixture was centrifuged at 15,000 rpm for 20 min, the supernatant was discarded, and the residue was suspended in 90 μL of 1× PBS to repeat the positive round process.

The eluted samples were amplified by PCR in 50 μL reactions, each containing Pfu Plus DNA polymerase and Pfu plus DNA polymerase Reaction Buffer (pH), 200 μM dNTP, and 0.5 μM forward and reverse primers. The PCR conditions were as follows: 95°C for 5 min, followed by 40 cycles of 95°C for 20s, 57°C for 20s, 72°C for 10s, and a final extension step of 5 min at 72°C. The PCR products were checked by electrophoresis on a 3% agarose gel, and the double-stranded DNA was purified by Expin CleanUp SV (GeneAll, Seoul, Korea). The concentration was determined by an Eppendorf BioSpectrometer basic. Finally, the purified dsDNA was separated into single-stranded DNA using Lambda Exonuclease. After incubation for 30 min at 37°C and deactivation for 10 min at 80°C, the ssDNA was produced, dissolved in 1× PBS buffer, and used for the next selection round after purification.

#### 5.2.2. Cloning and Sequencing of Selected DNA

After round 9, ssDNA pools selected by the GO-SELEX were amplified with Ligation Independent Cloning (LIC) primers. The purified PCR products were cloned using the LIC vector into DH5α cells (Enzynomics). Positive clones were selected and sequenced by Macrogen (Seoul, Korea). The secondary structure of the selected ssDNA sequences was predicted using the M-fold software.

### 5.3. Characterization of the Developed Aptamer

#### 5.3.1. CD Spectrum Measurement

The CD spectra were obtained to observe the conformational change of the aptamer caused by the binding to the target molecule using a CD spectrometer (J-815, JASCO Inc., Oklahoma City, OK, USA) [2,3]. Each sample, including the aptamer and the aptamer/STX, was prepared in the working buffer. The sample volume was adjusted to 400 μL and the concentrations of STX and the aptamer were fixed at 3 μM and 1 μM, respectively. The scan range and scan speed were adjusted to 190−350 nm and 200 nm/min, respectively. Triplicate measurements were obtained from each sample under flowing N_2_ gas and the wavelength interval was fixed at 0.2 nm.

#### 5.3.2. Fluorescence Assay

Fluorescence assay was performed to calculate the affinity of the aptamer [31]. The FAM-labeled aptamers were dissolved in the working buffer, heated at 90℃ for 3 min, and immediately cooled in an ice-bath for another 5 min. After stabilization for 15 min at room temperature (RT), the sample volume was adjusted to 300 μL and 10 to 150 nM of final concentration of the aptamer were incubated with 600 μg/L of STX at 37°C for 1 h in the dark. Subsequently, GO with an equal mass ratio to the aptamer was added to the sample and incubated at RT for 30 min. After centrifugation (15,000 rpm, 20 min), 200 μL of the supernatant containing STX-aptamer complex was transferred into a 96-well, flat-bottomed, black polystyrene microplate. In the case of the negative control, the same concentrations of the aptamer and GO were used without the target. The fluorescence intensities of the supernatants in the microplate were read by Cytation 5 Cell imaging Multi-Mod Reader (BioTek, winooski, VT, USA). The OriginPro2017 was used to plot the binding saturation curve and to calculate the dissociation constants (K_d_) through non-linear Hill equation fitting of the data.

### 5.4. Dvelopment of LSPR Aptasensor Chip

#### 5.4.1. Synthesis of GNRs

GNRs were synthesized using a seed-mediated growth procedure [44,45]. Briefly, 0.3 mL of 0.01 M NaBH_4_ was added to a seed solution which is comprised of 3.75 mL of 0.1 M CTAB and 0.12 mL of 0.01 M HAuCl_4_. The growth solution was prepared by mixing 4.72 mL of 0.1 M CTAB, 0.2 mL of 0.01 M HAuCl_4_, 0.025 mL of 0.01 M AgNO_3_, and 0.04 mL of 0.1 M ascorbic acid. Then, 40 μL of the seed solution was added to the growth solution and the resulting solution was incubated for 4 h at 30°C. Subsequently, the GNR was centrifuged for 20 min at 12,000 rpm and the pellet was dispersed in deionized water (DW) to remove excess CTAB. The centrifugation/resuspension procedure was repeated twice. The GNR synthesized using a seed-mediated method exhibited an average aspect ratio of 3.78 (35.6 ± 2.81 nm length and 9.4 ± 1.33 nm width). The size and aspect ratio were measured using high-resolution transmission electron microscopy (HR-TEM, JEM-2100, JEOL Ltd., Japan) (Appendix A).

#### 5.4.2. Fabrication of the GNR Substrate

Glass slides (15 × 10 × 0.15 mm) were cleaned with piranha solution (sulfuric acid:hydrogen peroxide = 3:1) at 65°C for 30 min, rinsed with DW and ethanol, immersed in an ethanolic solution of 2% (3-Mercaptoprophyl)trimethoxysilane (MPTMS) (Sigma-Aldrich, cat# 175617) at room temperature for 1 h and 30 min, and rinsed with ethanol and DW. Finally, the thiol-modified slide was exposed to a solution of GNR for 12 h. The presence of immobilized GNRs on the glass slide was confirmed using field emission scanning electron microscopy (FE-SEM, S-4700, Hitachi Ltd., Tokyo, Japan) and absorption spectrum analysis (Appendix A).

#### 5.4.3. Functionalization of GNR Substrate with Aptamers

The disulfide bonds between the STX aptamer and poly T_3_ (mixed with 1:5 molar ratio) were cleaved using 5 mM TCEP. To immobilize the STX aptamer and poly T_3_ on the GNR substrate, 100 μL of acetate buffer (pH 5.6, 10 mM) containing the mixture was loaded on the chip for 30 min at RT. Unbound aptamers were washed five times with deionized water (DW).

### 5.5. Determination of the Sensitivity and Specificity of the LSPR Aptasensor

The STX aptamer-coated GNR substrate was reacted in 100 μL of working buffer solution containing different concentrations of STX for 30 min at RT. Afterwards, the LSPR sensing chips were fixed to a transparent 96-well plate cover. Then, the absorption spectra were recorded from 650 to 850 nm wavelength at each reaction step using a 96-well plate reader (Cytation 5, BioTek, winooski, VT, USA). All the measured LSPR spectra were fitted with the OriginPro2017 program. The longitudinal absorption peak was fitted using a smoothing and non-linear Gaussian fitting process to calculate the maximum values of absorption bands. The fitted spectra were normalized to compare the maximum absorption wavelength.

### 5.6. Preparation of STX-Spiked Real Samples

Mussels were purchased from an online market. The mussel samples were homogenized in a blender. The extracts from the mussel samples were prepared following the official method of the AOAC [46]. An aliquot of 1.0 g of the homogenized sample was weighed into a 10-mL tube and samples were spiked with the STX standard solution (final concentrations of 10, 100, and 2000 μg/L). Then, 1 mL of 0.1 M hydrochloric acid was added to the tube and vortexed for 3 min. After extraction, the suspension was boiled at 95°C for 5 min and cooled to RT. After cooling, samples were centrifuged at 4500 rpm for 10 min, twice. Then, the supernatant was separated and filtered using a syringe filter (0.20 μm pore size). The filtrate was then analyzed after 1/100 dilution with the working buffer.

### 5.7. Statistical Analysis

In order to confirm reproducibility, all the LSPR signal measurements in the manuscript were performed six times. A total of six measured signals were utilized in this study to obtain reliable results with average, SD, and relative SD (RSD) values in optimized conditions, concentration patterns with standard/real samples and selectivity of the developed sensing system.

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
