# Peer review of "Label-Free Direct Detection of Saxitoxin Based on a Localized Surface Plasmon Resonance Aptasensor"

_toxins, 2019, doi:10.3390/toxins11050274_

Round 1

Reviewer 1 Report

Dear Authors,

After the review process, I have several comments: 

- you should eliminate results or data interpretation from the last paragraph of the introduction; 

- you should insert statistical data in figure 1A; 

- you should insert statistical section, for example 4.7; 

- you should present a limitation of the study; 

- you should present statistical relevance of the results, e.g. line 174; 

- you should eliminate discussion and Table 2 from the section 3 - I recommend to insert a Discussion section.

Best regards!

Author Response

Please see attached file, "Resoponses to Reviewers" document.

Thank you for your advices.

Reviewer 2 Report

The manuscript describes the development of an aptamer that binds the marine toxin saxitoxin (STX) and its application to detection of STX with a localized surface plasmon resonance (LSPR) sensor. The manuscript is generally well written and the quality of the experimental data is good. While several STX-specific aptamers have been described in the literature, this is apparently the first to be combined with LSPR.

General Comments:

It is clear from the lack of cross-reaction to gonyautoxins and neo-STX that the sensor has good selectivity for STX. However, I have several concerns about the sensitivity of the LSPR method and how it is described in the manuscript. The limit of detection (LOD) is given as 2.41 ppb. There are several problems with this. (1) The easiest to fix is the units (use ug/kg or ug/L instead). (2) The greater concern is that the method by which the LOD was measured was not described. For many sensors the LOD is defined as the concentration of analyte that will cause a change in signal three standard deviations from the mean of the analyte-free sample. If one looks at the calibration curve for STX by LSPR (figure 3B), by this metric the LOD looks like it would be in the range of 100-150 ppb. In fact, in Figure 3B it does not appear that the 20 ppb standard would be significantly different from the 2 ppb standard. (3) The legend in Figure 3 should also indicate if these experiments represent responses in buffer or in mussel extract.

Specific Comments:

The abstract should indicate the levels at which the mussel samples were spiked.

l.92: please add a citation for the work of professor Gu.

Scheme 1. This is a very nice graphic. It would benefit by increasing the font size of the smallest type used in it.

l.120: please define “FAM”

l.127-128. How much STX and how much aptamer were used?

Figure 1. This also is a nice figure, but the panels are too small to clearly see the features, especially for panels “B” and “D”.

l.138: replace “R” with the appropriate citation number.

Figure 2. (1) as with the other figures, the font is too small. (2) What the authors mean by “ratios” should be described in the legend. Is this the ratio of aptamer to spacer ? Also please describe what is meant by “OFF” and “ON”. This will save casual readers from having to hunt for this information in the manuscript.

l.180-185. The wording may need to be changed here. This appears to suggest that the shift is being caused by the cations. Is that true, or is this the effect of cations on a shift caused by STX? Please clarify.

Table 1. Please clarify how the recoveries in this table were calculated. For example, when the spiking level was 100 ppb, the amount found was 223.73 ppb. This would appear to indicate a “recovery” of 223%, not the 118.61% reported. Also, take care with the significant digits. Are the recovery values really accurate to 0.01%? Since the RSDs ranged from 2.5 to 3.5% there are not this many significant digits. More information on how the recoveries were calculated would be helpful.

l.230-231. There is a big difference between a sensor working on a few spiked samples in a laboratory, and being able to be used for “on-site” determination. What if the site doesn’t have electricity?

l.246: AOAC is not a regulatory authority. They are not affiliated with any government.

Table2. (1) Are the LODs reported in this table for the various methods in buffer or in matrix? (2) Is reference 19 really an electronic aptasensor method? (3) There are no “measurement steps” listed for the HILIC-MS/MS and LC-ESI-MS/MS methods. Do they not have sample cleanups or even simple sample dilution?

Author Response

Please see attached file, "Responses to Reviewers" document.

Thank you for your advices.

Round 2

Reviewer 1 Report

Dear Authors, I do not have any supplementary comments. Best regards!

Author Response

Thanks to your comments, the manuscript could be sophisticated. The advices you mentioned in major revision were great helpful. Thank you so much.

Reviewer 2 Report

The authors have addressed the concerns raised in the first review. The authors have justified their calculation of the LOD, which was one of the major concerns. However, the revision did raise another question. It appears that measurements were performed 6 times and the smallest and largest signals were discarded and the remaining 4 signals were used to calculate the averages and standard deviations (i.e. lines 388-392 in the revised manuscript). I am not a statistician, so I will not claim to know if this is acceptable or not, but it would seem that it would bias the data and reduce the standard deviations. Because the LOD was based upon the standard deviations, it might also bias that calculation too. I suggest the journal consult a statistician for their opinion. Other than this, the revised manuscript is very good.

On a less important note: do the error bars in Figure 3B report +/- one standard deviation or something different?

Author Response

Thanks to your comments, the manuscript could be sophisticated. The advices you mentioned in major/minor revision were great helpful. We appreciate deeply about that.
